# In Vitro Influence of ZnO, CrZnO, RuZnO, and BaZnO Nanomaterials on Bacterial Growth

**DOI:** 10.3390/molecules27238309

**Published:** 2022-11-28

**Authors:** Emad M. Abdallah, Abueliz Modwi, Samiah H. Al-Mijalli, Afrah E. Mohammed, Hajo Idriss, Abdulkader Shaikh Omar, Mohamed Afifi, Ammar AL-Farga, Khang Wen Goh, Long Chiau Ming

**Affiliations:** 1Department of Science Laboratories, College of Science and Arts, Qassim University, Ar Rass 51921, Saudi Arabia; 2Department of Chemistry, College of Science and Arts, Qassim University, Ar Rass 51921, Saudi Arabia; 3Department of Biology, College of Science, Princess Nourah Bint Abdulrahman University, Riyadh 11671, Saudi Arabia; 4Department of Physics, College of Science, Imam Mohammad Ibn Saud Islamic University (IMSIU), Riyadh 11623, Saudi Arabia; 5Department of Biological Sciences, Faculty of Sciences, King Abdulaziz University, Jeddah 21589, Saudi Arabia; 6Najla Bint Saud Al Saud Center for Distinguished Research in Biotechnology, Jeddah 21589, Saudi Arabia; 7Department of Biochemistry, College of Sciences, University of Jeddah, Jeddah 21577, Saudi Arabia; 8Department of Biochemistry, Faculty of Veterinary Medicine, Zagazig University, Zagazig 44519, Egypt; 9Faculty of Data Science and Information Technology, INTI International University, Nilai 71800, Malaysia; 10Pengiran Anak Puteri Rashidah Sa’adatul Bolkiah Institute of Health Sciences, Universiti Brunei Darussalam, Gadong BE1410, Brunei

**Keywords:** nanoparticles, *Klebsiella pneumonia*, *Staphylococcus aureus*, *Escherichia coli*, *Pseudomonas aeruginosa*

## Abstract

In this work, ZnO, CrZnO, RuZnO, and BaZnO nanomaterials were synthesized and characterized in order to study their antibacterial activity. The agar well diffusion, minimum inhibitory concentration (MIC), and minimum bactericidal concentration (MBC) assays were used to determine the antibacterial activity of the fabricated nanomaterials against *Staphylococcus aureus* ATCC 29213, *Escherichia coli* ATCC35218, *Klebsiella pneumoniae* ATCC 7000603, and *Pseudomonas aeruginosa* ATCC 278533. The well-diffusion test revealed significant antibacterial activity against all investigated bacteria when compared to vancomycin at a concentration of 1 mg/mL. The most susceptible bacteria to BaZnO, RuZnO, and CrZnO were *Staphylococcus aureus* (15.5 ± 0.5 mm), *Pseudomonas aeruginosa* (19.2 ± 0.5 mm), and *Pseudomonas aeruginosa* (19.7 ± 0.5), respectively. The MIC values indicated that they were in the range of 0.02 to 0.2 mg/mL. The MBC values showed that the tested bacteria’s growth could be inhibited at concentrations ranging from 0.2 to 2.0 mg/mL. According to the MBC/MIC ratio, BaZnO, RuZnO, and CrZnO exhibit bacteriostatic effects and may target bacterial protein synthesis based on the results of the tolerance test. This study shows the efficacy of the above-mentioned nanoparticles on bacterial growth. Further biotechnological and toxicological studies on the nanoparticles fabricated here are recommended to benefit from these findings.

## 1. Introduction

Over the last decade, nanostructure materials have been intensely studied because of their distinctive physicochemical and biological characteristics [1]. There has been tremendous scientific interest in nanomaterial applications, including for medical, engineering, agricultural, and other technological applications [2]. Scientists have attempted to encapsulate pharmaceuticals in virus-sized nanoparticles (NPs), and it is expected that in the near future, these nanoparticles will appear to be beneficial in gene therapy and cancer treatment [3]. Currently, there is an increasing interest in developing novel antibacterial agents to use as alternatives to conventional antibiotics, which are fast becoming ineffective in the face of bacteria that have developed high resistance rates [4,5]. Even though nanomaterials are created from the same polymer, their behavior might vary based on the underlying features that come from the preparation process. The primary features are surface density, diameter, permeability, nanomaterial density and photosensitizer performance. Due to these characteristics, the morphology of materials varies, which is predicted to impact their ultimate qualities. Several studies dealing with microbes have shown the effectiveness of nanofibrous materials as an antimicrobial barrier or agent [6,7]. In addition, due to their economic viability, excellent chemical reactivity, and electrical conductivity, several man-made nanoparticles are suggested for the treatment of wastewater from microbial and chemical waste. This may substitute fresh water for irrigation, industrial operations, or recreational activities [8].

Despite the plethora of uses for nanoparticles, little is known about the basic interactions between fabricated nanomaterials and bacteria [9]. NPs are being increasingly employed to target bacteria and this new technology might be very useful in the treatment of bacterial infections. Antibacterial coatings can be used on medical implants and process materials to keep them from getting infected and to aid in wound healing. Inorganic nanomaterials coated with porphyrin photosensitizers are being used in new antibiotic therapies. When these nanoparticles are exposed to visible light, they make singlet oxygen, which kills bacteria in the area [10]. Inorganic nanoparticles can also be used to transport antibiotics, develop microbial diagnostic tests, and innovate antibacterial vaccines to fight infectious diseases [11,12,13,14]. Additional advances and investigations are required to convert the idea of nanoparticle technology into one with a feasible, practical use as the next generation of medication delivery systems [15]. Further knowledge of the various processes behind the biological interactions of antibacterial nanomaterials as well as particle engineering is necessary. Zinc oxide nanomaterials and their derivatives have excellent characteristics as antibacterial agents. Therefore, the current work sought to assess the antibacterial activity of the synthesized nanomaterials against *Staphylococcus aureus* ATCC 29213, *Escherichia coli* ATCC35218, *Klebsiella pneumoniae* ATCC 7000603, and *Pseudomonas aeruginosa* ATCC 278533.

## 2. Materials and Methods

### 2.1. Fabrication of CrZnO, RuZnO, and BaZnO Nanomaterials

A total of 100 mL of 0.03 M zinc citrate dihydrate (Zn (NO_3_)_2_.2H_2_O) solution and 0.64 g of RuCl_3,_ BaCl_2_ or CrCl_3_ were added to the required amount of pectinose solution in a 1000 mL beaker at 270 °C, which was stirred for 3 h. After powder production, it was cooled and dried for 20 h at room temperature. Then, once the white powder had dried, it was annealed at 500 °C for 3 h to eliminate carbon, which resulted in an insignificant result. In addition, the nano-powders were thoroughly evaluated in terms of their structural, morphological, and chemical bonding and their antibacterial activity by employing several types of bacteria.

### 2.2. Characterizations Nanomaterials

Through X-ray diffraction (XRD), the crystal structure, phase purity, crystalline size, and lattice parameters of pure ZnO, Cr-doped, and Ru-doped ZnO nanomaterials were determined. JEOL’s field emission scanning electron microscopy (FESEM) was utilized to analyze the nanoparticles’ morphology. The energy dispersive X-ray spectrometer (EDS) examination was performed to determine the nominal stoichiometry composition of all sample surfaces. The nanoparticles’ chemical bonding and vibration modes were recorded using Fourier Transform Infrared (FTIR) spectra (JASCO FI-IR 460 spectrometer) between 400 and 4000 cm^−1^.

### 2.3. Bacterial Cultures

In the current study, one gram-positive, *Staphylococcus aureus* ATCC 29213, and three gram-negative bacteria, *Escherichia coli* ATCC 35218, *Klebsiella pneumoniae* ATCC 7000603, and *Pseudomonas aeruginosa* ATCC 278533, were obtained from the Bio-house medical laboratory in Riyadh, Saudi Arabia. The agar well diffusion assay was used to determine the antibacterial activity. Subcultures of each strain were grown on nutrient agar medium (Oxoid) plates for 24 h at 37 °C. Following that, a 0.5 McFarland standard was used to adjust the bacterial suspensions to about 1.5 × 10^8^ CFU/mL for well-diffusion, MIC, and MBC tests.

### 2.4. Agar Well-Diffusion Method

The agar well diffusion method was used to perform the antibacterial susceptibility testing. The antibacterial activity of synthesized nanoparticles was evaluated against four bacterial strains: one Gram-positive bacteria, *Staphylococcus aureus* (*S. aureus*), and three Gram-negative bacteria, *Klebsiella pneumoniae* (*K. pneumoniae*), *Pseudomonas aeruginosa* (*P. aeruginosa*), and *Escherichia coli* (*E. coli*). Bacteria were cultivated in Riyadh’s Bio-house medical laboratory. Each strain was sub-cultured on nutrient agar medium (Oxoid) and plates were incubated at 37 °C for 24 h. Bacterial suspensions in normal saline (0.9%) were prepared using the direct colony suspension technique and adjusted to the 0.5 McFarland standard to get a concentration of 1.5 × 10^8^ CFU/mL. The plates were inoculated with the tested bacterial strains, and 40 mL of the synthesized nanoparticles diluted in DMSO (1 mg/mL) was individually placed into each well of the Petri plates and left to dry for 1 h under aseptic conditions. As the negative control, DMSO was used, antibiotic (vancomycin, 1 mg/mL) was used as the positive control. And plates were incubated at 37 °C for 24 h according to the Clinical and Laboratory Standards Institute. The clean area around the well was measured and represented as a width in millimeters [16].

### 2.5. Minimum Inhibitory Concentration (MIC) Test

This test was performed to determine the lowest concentration of antibacterial agents at which about 90% of the bacterial growth is inhibited. The MIC was estimated for synthesized nanoparticles against the four tested bacterial strains *E. coli*, *S. aureus*, *K. pneumoniae,* and *P. aeruginosa*. A bacterial suspension of 300 μL at a concentration of 0.5 McFarland (1.5 × 10^8^ CFU/mL) was seeded into each tube of 9 mL nutrient broth individually, following the addition of 1 mL of different concentrations of synthesized nanoparticles at 2, 0.2, 0.02, 0.002, 0.0002 mg/mL and DMSO, which was used as the negative control, then incubated at 37 °C overnight. The values for the MIC have been recorded by comparing tube turbidity using the naked eye with that of a 0.5 McFarland standard medium. The concentration in the transparent tube, which indicated the growth prevention by antibacterial agents, is considered as the MIC values [17].

### 2.6. Minimum Bactericidal Concentration (MBC) Test

This test was conducted to determine the lowest antibacterial agent concentration at which approximately 100% of bacterial cells are killed. MBC was obtained by sub-culturing 50 μL of each MIC tube’s test dilution on nutrient agar plates. MBC value was defined as the highest dilution that indicated no single bacterial colony. Additionally, the MBC/MIC ratio was computed [18].

### 2.7. Statistical Analysis

All experiments were replicated at least three times independently, and the results were analyzed using the one-way ANOVA test with a significance level of *p* ≤ 0.05.

## 3. Results and Discussion

### 3.1. Structural Report of Nanomaterials

The scientific community has recently focused on ZnO nanostructures due to their biocompatibility, in order to investigate and understand their cytotoxic effects, interactions with biological systems such as nucleic acids, proteins, fats, tissues, cell membranes, biological fluids, etc., and biosafety for the effective usage in the biomedical field [19]. Figure 1 illustrates the X-ray diffraction (XRD) patterns for the pure and Cr-doped ZnO, Ru-doped ZnO, and Ba-doped ZnO samples. X-ray diffraction patterns of pure, Ru-doped and Cr-doped ZnO powders were analyzed to evaluate the influence of Ru and Cr on the structure and size of features of the ZnO. Figure 1 depicts the diffraction patterns for pure ZnO, Cr-doped ZnO, and Ru-doped ZnO samples. From the given figure, seven prominent peaks at 31.7°, 34.6°, 364°, 47.8°, 57.1°, 62.5°, and 67.3° were observed, which might be ascribed to diffraction from the (100), (002), (101), (102), (110), (103), and (112) planes, respectively. The structural properties of ZnO, CrZnO, BaZnO, and RuZnO nanomaterials are presented in Table 1. The pure ZnO nanoparticle feature shows a hexagonal wurtzite phase crystal structure in the XRD pattern (JCPDS: 36-1451) [20]. The researcher documented that the insertion of dopant materials changes the lattice features of the host materials [21]. A Ru dopant in the ZnO crystal lattice does not produce any additional secondary phases. Similarly, no changes in the XRD pattern of Cr-doped ZnO NPs are seen in the diffraction data. It is worth noting that the intensity of the XRD peak reduces in Cr-doped ZnO, but it increases in Ru-doped ZnO. This could be attributed to the larger ionic radius of Cr (0.063 nm) compared to the Ru ionic radius (0.062 nm), hence a decrease in the XRD peak would be expected. Additionally, sharp or powerful peaks suggest a higher level of order, crystallinity, and configuration. By contrast, the reduction in peak intensity exhibits a lower level of crystallinity [22,23]. The nanomaterials’ average crystalline size (D nm) was calculated using the famous Scherrer formula from the top peak (101), and the results are tabulated in Table 1 [24].

### 3.2. Morphology and Chemical Composition

The FESEM photography approach is used to examine the shape and aggregation of the produced nanoparticles. SEM photographs of ZnO, CrZnO, BaZnO, and RuZnO nanoparticles and their elemental composition are shown in Figure 2. The obtained pictures confirm the formation of nanoparticle materials. The pure Zinc Oxide nanoparticles exhibit oval morphology distributed regularly without any impurities, as shown in Figure 2a and the EDX plot (Figure 2a,b). The picture shows that the nanoparticles are quite small, which is consistent with the particle sizes computed using the Debye–Scherrer formula. The particle sizes computed from the TEM images were found to be in the nanoscale range 33–80, which is compatible with the XRD data; this implies the formation of nanoparticle materials. When ZnO is doped with chromium, a cluster forms, causing the crystals to grow larger, confirming the irregular distribution and direction of the nanoparticles from oval to undifferentiated morphology. In contrast, doping ZnO with Ru results in an evenly distributed oval, with Ru pieces arranged at short distances on the surface of certain ZnO surface crystals. Ru-doped ZnO has well-defined morphological characteristics, with less agglomeration and a more uniform particle size distribution than Cr-doped ZnO. In the scientific literature, N-doped ZnO films have been created by utilizing a variety of deposition processes, including sputtering, pulsed laser deposition, spray pyrolysis, chemical vapor deposition, and implantation, all of which use diverse nitrogen sources. However, there is ongoing debate over the repeatability and dependability of generating p-type ZnO:N [25].

### 3.3. FTIR Analysis of Fabricated Nanomaterials

Figure 3 represents the FTIR spectra observed in 500–4000 cm^−1^ for ZnO, RuZnO, CrZnO, and BaZnO nanomaterial. The wide band at 3378–3395 cm^−1^ referred to the O–H stretching vibration of the acetate -COOH precursor and adsorbed H_2_O humidity [26]. The Zn–O bonding matches the 482 cm^−1^ stretching wavelengths, verifying the production of ZnO. IR bands associated with O–H bonding at 1451 cm^−1^ demonstrate a greater transmission level in BaZnO, which might be related to the difference in Zn^+2^ and Ba^+2^ radii as well as the structural changes caused by doping. The measured peak at 2346 cm^−1^ is a result of the absorption of CO_2_ in the environment, which exhibit C=O stretching vibrations [27]. Because of the formation of a Zn–Ba–O connection, the peak seems unusually broad, reaching 573 cm^−1^ [28]. The two intense bands at 1436 and 1077 cm^−1^ in CrZnO can be equated to the vibration of the Zn–Cr bonds, demonstrating the effective incorporation of Cr ions into ZnO. Furthermore, the bands at 881 cm^−1^ might represent the stretching vibration of Cr-O. Thus, the production of Cr-O implies that Cr atoms have been incorporated into the ZnO lattice. RuZnO has a prominent peak at 484 cm^−1^ in its FTIR spectrum, which is likewise related to Zn-O stretching vibrations [29]. FTIR study findings of CrZnO, RuZnO, and BaZnO nanomaterials are consistent with XRD pattern findings. The findings established in the FTIR analysis concur with XRD analysis.

### 3.4. Antibacterial Potential

Nanotechnology has recently emerged as one of the most significant technologies for potential antibacterial chemotherapy, owing to its precise targeting, which is associated with a reduction in the dosages needed and, as a result, fewer predicted medication side effects [30]. The antibacterial activity of the investigated nanomaterials (BaZnO, RuZnO, and CrZnO) at a concentration of 1 mg/mL was tested against the referenced bacterial strains representing gram-positive and gram-negative bacteria using the agar well-diffusion technique. Standard antibiotics, vancomycin (1 mg/mL), and ZnO nanoparticles were selected as the control group (at the same dose). The zone of inhibition of the examined bacteria with DMSO (negative control) was not detected. The results demonstrated that the fabricated nanomaterials (BaZnO, RuZnO, and CrZnO) had significant antibacterial activity at 1 mg/mL (*p* ≤ 0.05) when compared to the antibiotic and ZnO nanoparticles. Generally, the mean zone of inhibition of the ZnO nanomaterial (the control) ranged between 12.2 ± 0.9 mm and 19.7 ± 0.5 mm, and the most susceptible bacterium was *S. aureus*; vancomycin, at the same concentration, showed inhibition zones ranging between 12.5 ± 0.5 and 15.5 ± 0.5, and the most susceptible bacterium was *E. coli*. For the fabricated nanomaterials, the BaZnO nanomaterial inhibition zones ranged from 8.7 ± 0.5 mm to 15.7 ± 0.5 mm, and the most susceptible bacterium was *S. aureus*. The RuZnO nanomaterial inhibition zones ranged from 10.7 ± 0.9 mm to 19.2 ± 0.5 mm, and the most susceptible bacterium was *P. aeruginosa*. The CrZnO nanomaterial inhibition zones ranged from 12.2 ± 0.5 mm to 19.2 ± 0.5 mm, and *P. aeruginosa* was the most sensitive bacterium (Figure 4).

The agar well-diffusion test was selected for this study based on published findings that the inhibition zone is larger in well-diffusion measurements than in other assays such as disk diffusion. This is because the well-diffusion test doubles the volume of nanomaterial suspensions and increases nanomaterial diffusion through the medium, resulting in increased antibacterial activity [30,31]. Previous studies have shown that ZnO nanoparticles have a wide range of antibacterial abilities against various bacterial species, and that ZnO nanoparticles have significantly more antibacterial abilities against *Staphylococcus aureus* in particular [32,33]. To our knowledge, no data on the antibacterial activity of the nanomaterials fabricated in this investigation (BaZnO, RuZnO, and CrZnO) has been reported. However, various publications cited that some nanomaterials doped in ZnO nanostructure showed noticeable antibacterial activity [34,35,36,37,38,39,40]. Table 2 highlights some of these previous reports in comparison to our results. Gram-positive and Gram-negative bacteria responded well to ZnO nanoparticles’ antibacterial properties, and the particle size has a significant impact on ZnO’s broad-spectrum antibacterial activity [41]. According to studies, ZnO’s antibacterial activity increases as the particle size decreases. The smallest ZnO NPs (>1 m, 8 nm) had more antibacterial activity than MgO, TiO_2_, Al_2_O_3_, CuO, and CeO_2_ NPs [42].

The MIC, MBC, and MBC/MIC values are shown in Table 2. Notably, most MIC values were 0.2 mg/mL, indicating that these nanoparticles have potential bacteriostatic effects against most tested bacteria at a concentration of 0.2 mg/mL, while the majority of MBC values were 2.0 mg/mL. Since the MIC values of these nanomaterials were very low and the MBC values were rather high (values ranged between 10.0 and 100.0), we determined that their inhibitory effect was bacteriostatic (MBC/MIC ≥ 4.0) against all tested bacteria at 1 mg/mL. The MBC/MIC ratio is a metric that indicates a compound’s bactericidal capability by comparing the two values: when the ratio of MBC/MIC is less than or equal to four, the sample is bactericidal, and when the ratio is more than four, the sample is bacteriostatic [43]. There is a widespread perception that bactericidal antibacterial agents are more effective than static antibacterial agents, but there is no scientific evidence supporting this. Both “cidal” and “static” are words used in the laboratory that relate to the impact of antibacterial drug concentrations on bacterial growth over a specified tolerance. They are unable to forecast the fate of an infection in vivo. Most antibacterial agents directed against the bacterial cell wall are bactericidal, while those targeted at bacterial protein synthesis are mostly bacteriostatic [44]. For instance, beta-lactam antibiotics, such as cephalosporins, are bactericidal drugs that prevent or stop the synthesis of bacterial cell walls. Conversely, bacteriostatic antibiotics such as chloramphenicol and clindamycin act by inhibiting protein synthesis to reduce or prevent bacterial growth. The fundamental information on an antibacterial agent’s action is provided by the MIC and MBC tests, and Raman spectroscopy can shed more light on this process (bacteriostatic or bactericidal) [45].

Additionally, MBC/MIC ratios could also be a good indication for the tolerance levels assessment for the bacteria isolates against nanomaterials concentrations. The current investigation revealed MBC/MIC ≤ 32, and thus that the tested strains were not tolerant to the investigated nanomaterials [46]. Overall, our findings corroborate earlier literature on ZnO, in that the pure powdered nanoparticles of ZnO showed bacteriostatic action against some gram-negative and gram-positive bacteria, as shown in Table 3 [47]. On the other side, it was also mentioned that the fabricated TiO_2_ZnO nanoparticles showed a bactericidal effect against some gram-negative and gram-positive bacteria [48]. The bacteriostatic or bactericidal activity might be attributed to the nanoparticle concentration employed to eliminate microorganisms, given that the modification of ZnO nanoparticles with other nanomaterials alters the antibacterial properties of the nanoparticles as well [49]. Even though the antibacterial processes of NPs are not fully understood, the most common explanations include metal ion release, oxidative stress, and non-oxidative actions [50]. Thus, the intention of the MIC and MBC tests in the current study was to get a better understanding of the mode of action of these doped nanomaterials.

Furthermore, it is thought that the mode of action of antibacterial nanoparticles is distinct from that of conventional antibiotics. The antibacterial activity of ZnO nanoparticles and their derivatives is not well understood and remains a source of contention [51]. Direct contact of ZnO nanoparticles and their derivatives with bacterial cell walls, on the other hand is likely to result in the loss of bacterial cell viability and the release of antibacterial ions [52,53]. Additionally, it is also claimed that that these nanoparticles are capable of accumulating excessive reactive oxygen species (ROS) and metal ions, disrupting normal bacterial cell homeostasis [54]. The current study concludes that BaZnO, RuZnO, and CrZnO nanomaterials may penetrate the bacterial cell membrane and interact with the cell’s basic components such as DNA, ribosomes, or enzymes, leading to protein deactivation based on their potential bacteriostatic effects at 1 mg/mL. Future work on the antioxidant activity and antibacterial mode of action of these novel doped nanomaterials is recommended (Figure 5). Moreover, additional toxicological studies on the nanoparticles synthesized here are required, since several risks and toxicities associated with ZnO nanoparticles have been documented after systemic ZnO nanomaterial exposure [55]. Although several materials and nanoparticle platforms have been created, relatively few articles analyze the nanoparticle platform’s toxicity in detail. Therefore, the development of these nanomaterials has not been accompanied by a serious investigation of their long-term effects to allow the risk-to-benefit ratio to be balanced. Consequently, toxicology studies are needed on these novel nanomaterials [56].

## 4. Conclusions

In this investigation, ZnO, CrZnO, BaZnO, and RuZnO nanomaterials were fabricated by a co-deposition method and used as inhibitory materials for bacterial growth. Experiments demonstrate that the synthesized nanomaterials have remarkable antibacterial action against both gram-positive and gram-negative bacteria, suggesting that they can penetrate thin and thick peptidoglycan cell walls. Low MIC and MBC values were reported, indicating that the bacterium had high antibacterial ability. The antibacterial effect is attributed to their bacteriostatic characteristics, which target bacterial protein synthesis after penetrating the bacterial cell walls, according to the MBC/MIC ratio. Overall, the nanomaterials synthesized in the current study are suitable as models for use in biotechnological or medical applications because of their stability, durability, and antibacterial capabilities. However, for a better understanding and potential implementation of these synthesized nanoparticles in diverse antibacterial applications, further research is needed, such as long-term therapy and kinetics. For future uses in the pharmaceutical sector, it is essential to evaluate their effects, risks, and safety in relation to eukaryotic cells.

## Figures and Tables

**Figure 1 molecules-27-08309-f001:**
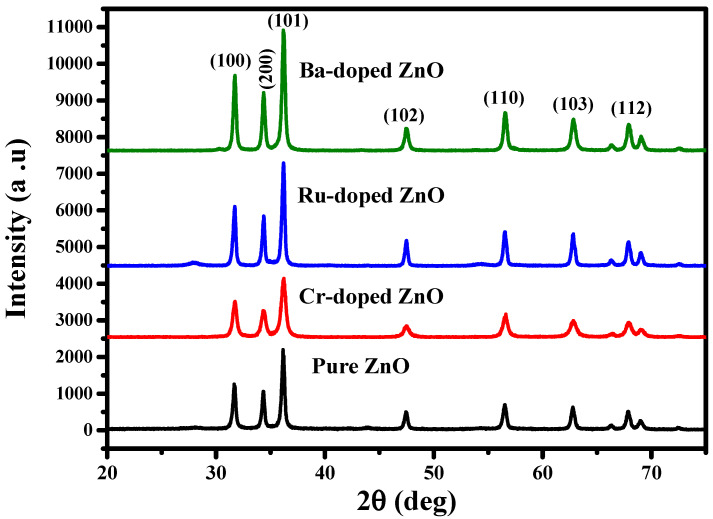
The XRD pattern of ZnO, CrZnO, RuZnO, and BaZnO nanomaterials.

**Figure 2 molecules-27-08309-f002:**
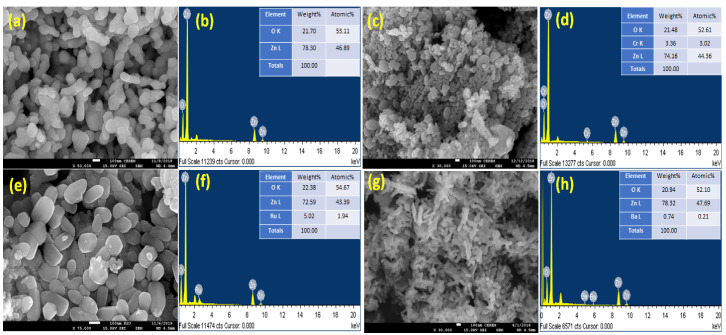
Morphology and chemical composition of ZnO, CrZnO, BaZnO, and RuZnO nanoparticles. (**a**,**b**): ZnO, (**c**,**d**): CrZnO, (**e**,**f**): RuZnO, (**g**,**h**): CrZnO.

**Figure 3 molecules-27-08309-f003:**
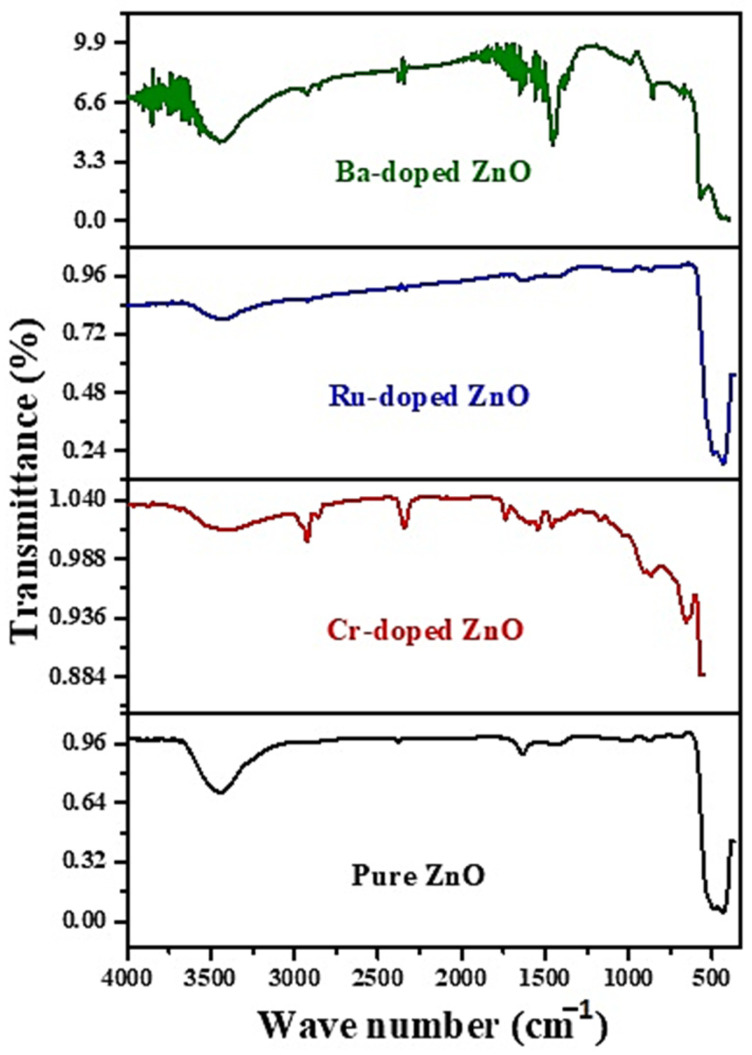
FTIR spectrum of the ZnO, CrZnO, RuZnO, and BaZnO nanomaterials.

**Figure 4 molecules-27-08309-f004:**
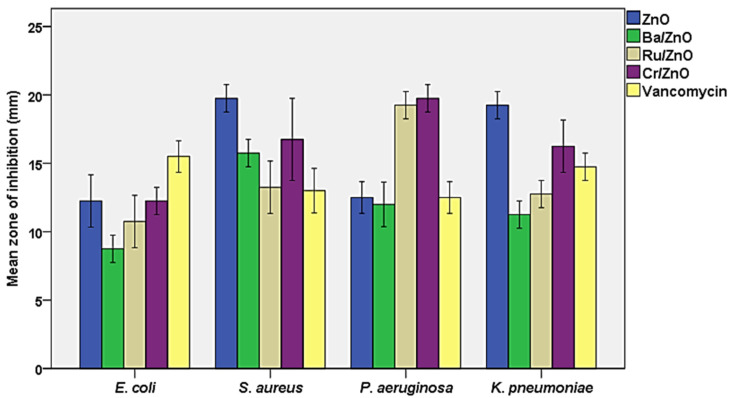
Susceptibility of the examined bacteria to the three fabricated nanomaterials compared to ZnO nanoparticles and vancomycin at (1 mg/mL).

**Figure 5 molecules-27-08309-f005:**
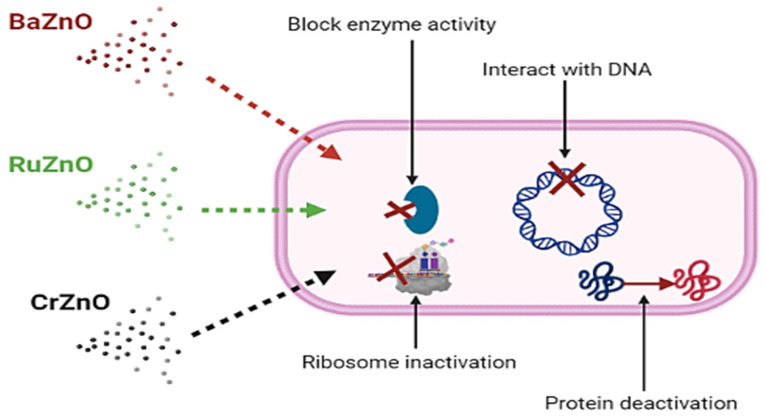
The possible mechanisms of the BaZnO, RuZnO, and CrZnO nanomaterials fabricated via the co-deposition method.

**Table 1 molecules-27-08309-t001:** Structural properties of the fabricated nanomaterials.

Samples	2θ (101)	FWHM	D (nm)	Lattice Parameters		c/a	d (Å)
				a (Å)	c (Å)		
Pure ZnO	36.19	0.263	33.20	3.254	5.214	1.602	2.482
Cr-doped ZnO	36.20	0.391	22.00	3.254	5.214	1.602	2.479
Ru-doped ZnO	36.18	0.254	34.38	3.252	5.210	1.602	2.482
Ba-doped ZnO	36.35	0.412	21.21	3.2537	5.217	1.603	2.469

**Table 2 molecules-27-08309-t002:** The diameter of the inhibition zone against different microorganisms in ZnO doped with various metallic elements.

Nanomaterial	Concentration	Zone of Inhibition	Gram-Positive	Gram-Negative	Reference
Ni/ZnO	2 mg/mL	18.0 0.0 14.0 11.0	*-* *-* *S. aureus* *S. epidermidis*	*E. coli* *A. baumannii* *-* *-*	[37]
Cu/ZnO	1 mg/mL	24.0 20.0 18.0 17.0	*S. aureus* *S. pyogenes* *-* *-*	*-* *-* *E. coli* *K. Pneumoniae*	[38]
Ag/ZnO	1 wt%	19.0	*S. aureus*	*-*	[39]
In-doped ZnO	3%	12.0 18.0 16.0 14.0	*S. aureus* *Bacillus subtilis* *-* *-*	*-* *-* *E. coli* *P. aeruginosa*	[40]
Ce/ZnO	0.4 mol %	16.0 0.0	*S. aureus* *-*	*-* *E. coli*	[41]
Ba/ZnO	1 mg/mL	15.5 8.7 11.2 12.0	*S. aureus* *-* *-* *-*	*-* *E. coli* *K. pneumoniae* *P. aeruginosa*	Present study
Ru/ZnO	1 mg/mL	13.2 10.7 12.7 19.2	*S. aureus* *-* *-* *-*	*-* *E. coli* *K. pneumoniae* *P. aeruginosa*	Present study
Cr/ZnO	1 mg/mL	16.7 12.2 16.2 19.7	*S. aureus* *-* *-* *-*	*-* *E. coli* *K. pneumoniae* *P. aeruginosa*	Present study

**Table 3 molecules-27-08309-t003:** The MIC, MBC, and MBC/MIC values for the nanomaterials.

Bacterial Strains	MICs (mg/mL)				MBCs (mg/mL)				MBC/MIC Values			
	ZnO	Ba/ ZnO	Ru/ ZnO	Cr/ ZnO	ZnO	Ba/ ZnO	Ru/ ZnO	Cr/ ZnO	ZnO	Ba/ ZnO	Ru/ ZnO	Cr/ ZnO
*E. coli*	0.2	0.2	0.2	0.2	2.0	2.0	2.0	2.0	10.0	10.0	10.0	10.0
*S. aureus*	0.02	0.2	0.2	0.2	0.2	2.0	2.0	2.0	10.0	10.0	10.0	10.0
*K.* *pneumoniae*	0.02	0.2	0.02	0.2	0.2	2.0	0.2	2.0	10.0	10.0	100	10.0
*P.* *aeruginosa*	0.2	0.2	0.2	0.02	2.0	2.0	2.0	0.2	10.0	10.0	10.0	100

## Data Availability

Not applicable.

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
