# Peer review of "In Vitro Influence of ZnO, CrZnO, RuZnO, and BaZnO Nanomaterials on Bacterial Growth"

_molecules, 2022, doi:10.3390/molecules27238309_

Round 1
Reviewer 1 Report
I have review the paper entitled “In vitro Influence of ZnO, Cr@ZnO, Ru@ZnO and Ba@ZnO 2 Nanomaterials on Bacterial Growth”. The paper contains very good data however needs some changes before acceptance.
Comments
-The abstract should be more informative and include key results in the abstract.
-The name of the bacterial strain in the abstract and the entire text should be in italics.
- The introduction section should be more updated.
-The formula of chemicals should be written in a proper format.
-The results and discussion part should be improved.
Author Response
We would like to thank the Reviewer for taking the time and effort necessary to review the manuscript. We sincerely appreciate all valuable comments and suggestions, which helped us to improve the quality of the manuscript, we confirm that the manuscript was proofread by an English native professional (Certificate sent to the editor), and we also followed and corrected the entire manuscript according to the reviewer's comments, following the comments point by point. Here are the replies to the comments of the reviewer:
I have review the paper entitled “In vitro Influence of ZnO, Cr@ZnO, Ru@ZnO and Ba@ZnO 2 Nanomaterials on Bacterial Growth”. The paper contains very good data however needs some changes before acceptance.
Comments
-The abstract should be more informative and include key results in the abstract. Done!
-The name of the bacterial strain in the abstract and the entire text should be in italics. Done!
- The introduction section should be more updated. Done!
-The formula of chemicals should be written in a proper format. Thank you; all the chemical formulas have been modified.
-The results and discussion part should be improved. Done!
Reviewer 2 Report
This manuscript (Title: In vitro Influence of ZnO, Cr@ZnO, Ru@ZnO and Ba@ZnO 2 Nanomaterials on Bacterial Growth) aims to study the antibacterial activity of nanomaterials. The science story is good and results exhibited sounds well.
(1) Line 147. “It is worth noting that the intensity of the XRD peak reduces in Cr-doped ZnO, but it increases in Ru-doped ZnO.” Could you please give some reasons about this result?
(2) Line 163. “The picture shows that the nanoparticles are quite small, which is consistent with the particle sizes computed using the Debye-Scherrer formula” Very interesting, could you please give more analysis?
(3) Line 246. “Most antibacterial agents directed against the bacterial cell wall are mostly bactericidal, while those targeted at bacterial protein synthesis are bacteriostatic” Could you please give more discussion on this mechanism?
(4) Line 265-280. This section is suggested to be represented by a mechanism diagram, which is more intuitive
Author Response
We would like to thank the Reviewer for taking the time and effort necessary to review the manuscript. We sincerely appreciate all valuable comments and suggestions, which helped us to improve the quality of the manuscript. We sent the manuscript to a native English language professional for proofreading (certificate sent to the editor). Regarding other comments, we followed up and added or corrected the required parts following the comments of the reviewer, point by point:
This manuscript (Title: In vitro Influence of ZnO, Cr@ZnO, Ru@ZnO and Ba@ZnO 2 Nanomaterials on Bacterial Growth) aims to study the antibacterial activity of nanomaterials. The science story is good and results exhibited sounds well. Thank you so much, we appreciate it.
(1) Line 147. “It is worth noting that the intensity of the XRD peak reduces in Cr-doped ZnO, but it increases in Ru-doped ZnO.” Could you please give some reasons about this result? Thank you; the reason was added in the text:
“This could be attributed to the larger ionic radius of Cr (0.063 nm) compared to Ru ionic radius (0.062nm); hence a decrease in the XRD peak is expected”.
(2) Line 163. “The picture shows that the nanoparticles are quite small, which is consistent with the particle sizes computed using the Debye-Scherrer formula” Very interesting, could you please give more analysis? Thanks, more analysis was added:
“The particle size computed from the TEM images was found to be in the nanoscale range of (33-80 nm), which is compatible with the XRD data; this implies the formation of nanoparticle materials”
(3) Line 246. “Most antibacterial agents directed against the bacterial cell wall are mostly bactericidal, while those targeted at bacterial protein synthesis are bacteriostatic” Could you please give more discussion on this mechanism? Thank you, more discussion added: “For instance, beta-lactam antibiotics, such as cephalosporins, are bactericidal drugs that prevent or stop the synthesis of bacterial cell walls. Contrarily, bacteriostatic antibiotics like chloramphenicol and clindamycin act by inhibiting protein synthesis to reduce or prevent bacterial growth. The fundamental information on the antibacterial agent's action is provided by the MIC and MBC tests, and Raman spectroscopy can shed more light on this process (bacteriostatic or bactericidal)".
(4) Line 265-280. This section is suggested to be represented by a mechanism diagram, which is more intuitive. Thank you so much for this excellent suggestion, A new figure was added to show the possible mechanism (Figure 5).